

# Integrated analysis of differentially expressed profiles and construction of a competing endogenous long non-coding RNA network in renal cell carcinoma

Qianwei Xing[*], Yeqing Huang[*], You Wu, Limin Ma and Bo Cai

Department of Urology, Affiliated Hospital of Nantong University, Nantong, Jiangsu, China
[*] These authors contributed equally to this work.

## ABSTRACT

**Background**. Long non-coding RNAs (lncRNAs) play crucial roles in the initiation and progression of renal cell carcinoma (RCC) by competing in binding to miRNAs, and related competitive endogenous RNA (ceRNA) networks have been constructed in several cancers. However, the coexpression network has been poorly explored in RCC.

**Methods**. We collected RCC RNA expression profile data and relevant clinical features from The Cancer Genome Atlas (TCGA). A cluster analysis was explored to show different lncRNA expression patterns. Gene ontology (GO), Kyoto Encyclopedia of Genes and Genomes (KEGG) pathway enrichment analyses and gene set enrichment analysis (GSEA) were performed to analyze the functions of the intersecting mRNAs. Targetscan and miRanda bioinformatics algorithms were used to predict potential relationships among RNAs. Univariate Cox proportional hazards regression was conducted to determine the RNA expression levels and survival times.

**Results**. Bioinformatics analysis revealed that the expression profiles of hundreds of aberrantly expressed lncRNAs, miRNAs, and mRNAs were significantly changed between different stages of tumors and non-tumor groups. By combining the data predicted by databases with intersection RNAs, a ceRNA network consisting of 106 lncRNAs, 26 miRNAs and 69 mRNAs was established. Additionally, a protein interaction network revealed the main hub nodes (VEGFA, NTRK2, DLG2, E2F2, MYB and RUNX1). Furthermore, 63 lncRNAs, four miRNAs and 31 mRNAs were significantly associated with overall survival.

**Conclusion**. Our results identified cancer-specific lncRNAs and constructed a ceRNA network for RCC. A survival analysis related to the RNAs revealed candidate biomarkers for further study in RCC.

## INTRODUCTION

Renal cell carcinoma (RCC) is among the ten leading cancer types for estimated new cancer cases in both males and females in the United States (*Siegel, Miller & Jemal, 2017*). Although the five-year relative survival rate for all cancers has increased due to advances in diagnosis and therapy over the past three decades, discriminative biological markers

Corresponding author
Bo Cai, caibo@ntu.edu.cn

have not been determined in RCC for early-stage diagnosis and therapy, unlike for other cancers. Increasing numbers of studies on RCC have been conducted to explore tumor initiation and progression, including protein-coding RNAs (mRNAs) and non-coding RNAs (ncRNAs) (*Frew & Moch, 2015*; *Lorenzen & Thum, 2016*; *Martens-Uzunova et al., 2014*; *Qin et al., 2014*; *Schmidt & Linehan, 2016*); however, the tumor-specific mechanisms in the regulation of tumor progression and biological behaviors are not fully understood. Recently, long non-coding RNAs (lncRNAs) have been reported to play important roles in the molecular mechanism of RCC (*Chen et al., 2016*; *Li, Shuch & Gerstein, 2017*; *Serghiou, Kyriakopoulou & Ioannidis, 2016*).

LncRNAs are a subtype of ncRNAs with transcript lengths over 200 nucleotides and were once viewed as transcriptional noise without the capacity to encode proteins (*Guttman et al., 2009*). However, growing evidence suggests that lncRNAs may play crucial biological roles in transcriptional regulation, cellular development, and RNA modification (*Schmitt & Chang, 2016*). Emerging reports have revealed that lncRNAs are associated with various human diseases, such as neurological diseases (*Bai et al., 2014*; *Lau, Frigerio & De Strooper, 2014*), cardiovascular diseases (*Vausort, Wagner & Devaux, 2014*; *Yang et al., 2014*), immune inflammatory diseases (*Kino et al., 2010*), and malignant tumors; moreover, lncRNAs can be drug targets (*Sana et al., 2012*). In urologic malignancies, lncRNAs are thought to be related to tumor formation, invasion, and metastasis (*Martens-Uzunova et al., 2014*). In RCC, several studies have reported that lncRNAs may function as oncogenes or tumor suppressors and that they can affect the long-term survival and mortality of patients. For example, TRIM52-AS1 overexpression can suppress cell migration and proliferation and can induce apoptosis of RCC (*Liu et al., 2016a*; *Liu et al., 2016b*). Additionally, down-regulated lncRNA H19 inhibits RCC carcinogenesis (*Wang et al., 2015*). Tumor formation and development are complex pathophysiological processes, and lncRNAs may interact with miRNAs or mRNAs or other molecules in regulatory capacities. There are multiple hypotheses regarding how cancer-specific molecules communicate with each other (*Gomes, Nolasco & Soares, 2013*; *Huang et al., 2017*).

In 2011, *Salmena et al. (2011)* proposed a competing endogenous RNA (ceRNA) hypothesis, which indicated that all types of RNA transcripts communicate with each other by competing for binding to miRNA response elements (MREs). This competition between mRNAs, lncRNAs, pseudogenes, and circular RNAs widely exists in tumor initiation and progression. *Xiao et al. (2015)* reported that the lncRNA MALATI participates in the ceRNA network by sponging miR-200s to regulate ZEB2 expression. Additionally, *Qu et al. (2016)* demonstrated that exosome-transmitted lncARSR acts as a ceRNA to promote sunitinib resistance in renal cancer. Nevertheless, studies incorporating large sample sizes and high-throughput detection methods to identify whether lncRNAs are correlated with overall survival, gender, or other clinical features or to determine the ceRNA mechanisms of lncRNAs with aberrant expression in RCC are currently lacking.

The Caner Genome Atlas (TCGA) (http://cancergenome.nih.gov) is a platform that can provide clinical information and RNA sequencing with miRNA, lncRNA, and mRNA data. We downloaded data sets from the TCGA that included RNA sequences of different stage (according to the American joint committee on cancer, AJCC) tumor tissues and

adjacent non-tumor renal tissues. To the best of our knowledge, this study is the first to use this large sequencing database to investigate the differential expression profiles of a cancer-specific lncRNA and ceRNA coexpression network in RCC. This new approach of predicting caner-specific lncRNAs and their ceRNA potential can help us elucidate the functions of lncRNAs in RCC.

## MATERIALS AND METHODS

### Patients and samples

A total of 537 patients with RCC were retrieved from the TCGA data portal. We selected data according to the following criteria: (1) the histopathological diagnosis was RCC; (2) complete data was required for all samples of all stages; and (3) the included patients had no other malignancies. Overall, a total of 457 RCC patients were enrolled in this study. Among these samples, we analyzed miRNA sequences from 64 adjacent non-tumor tissue samples and 213 tumor tissue samples (95 stage I samples, 27 stage II samples, 46 stage III samples, and 45 stage IV samples) and mRNA/lncRNA sequences from 65 adjacent non-tumor tissues and 454 tumor tissues (209 stage I samples, 51 stage II samples, 114 stage III samples, and 80 stage IV samples). This study was conducted in accordance with the publication guidelines provided by the TCGA (http://cancergenome.nih.gov/publications/publicationguidelines); therefore, further approval of the ethics committee was not required.

### RNA sequencing data sets and computational analysis

The RCC RNA expression profile data (level 3) and relevant clinical features were acquired from the TCGA data portal (through Feb, 2017). The lncRNA and mRNA expression profiles were normalized using the RNASeqV2 system provided by the TCGA database. The miRNA expression profiles, downloaded from the TCGA database, were obtained using the Illumina HiSeq 2000 miRNA sequencing platform (Illumina Inc., San Diego, CA, USA). No further normalization was needed, as the TCGA database already had normalized the lncRNA, miRNA, and mRNA expression profile data. Then, we compared differentially expressed RNAs at four levels, including stage I RCC patient tumor tissues vs. non-tumor renal tissues, AJCC stage II RCC patient tumor tissues vs. non-tumor renal tissues, AJCC stage III RCC patient tumor tissues vs. non-tumor renal tissues, and AJCC stage IV RCC patient tumor tissues vs. non-tumor renal tissues. In the next step, we collected intersecting lncRNAs ($P \leq 0.01$, FDR $\leq 0.01$, FC $\geq 2$), miRNAs ($P < 0.05$, FDR $< 0.05$, FC $\geq 2$) and mRNAs ($P \leq 0.01$, FDR $\leq 0.01$, FC $\geq 3$)for hierarchical clustering and further bioinformatics analysis. A flow chart was depicted to show the framework of this study (Fig. 1).

### GO, KEGG pathway analysis and GSEA

Gene ontology (GO) analysis contains three domains: biological processes, cellular components, and molecular functions. We used a GO database to analyze differentially expressed intersecting mRNAs (http://www.geneontology.org). The molecular functions of up- and down-regulated genes were identified. The potential functions of the aberrantly

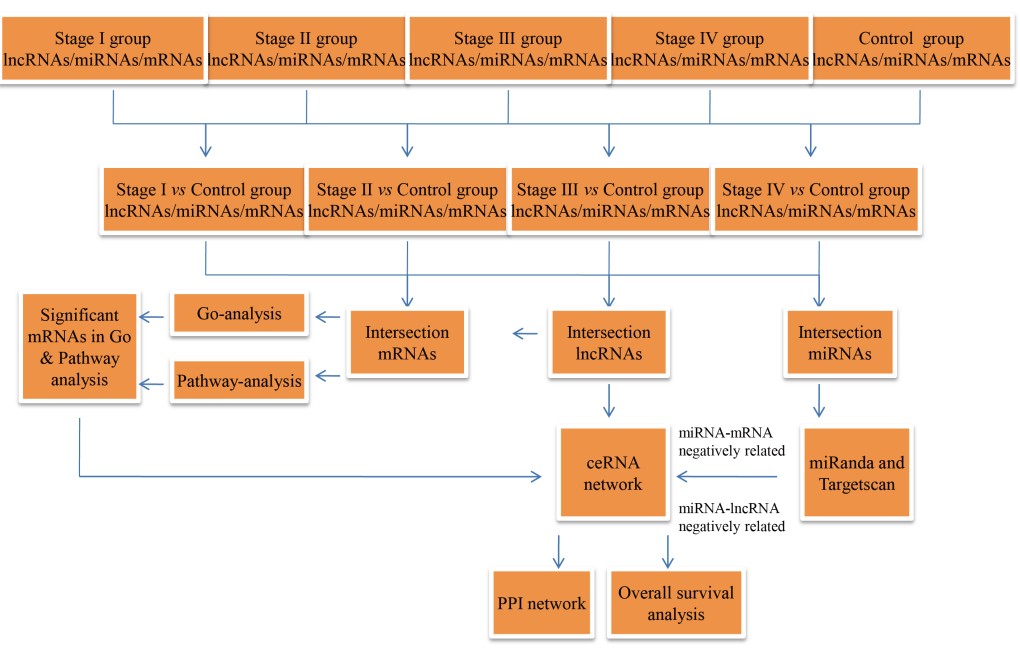

**Figure 1  Flow chart of the ceRNA network construction.**

expressed genes involved in the pathways were analyzed using the Kyoto Encyclopedia of Genes and Genomes (KEGG) (http://www.kegg.jp). Besides, the above data were analyzed using gene set enrichment analysis (GSEA) (Broad Institute, Cambridge, MA, USA, accessed at http://software.broadinstitute.org/gsea/). Additionally, we identified intersecting mRNAs ($P < 0.05$, FDR $< 0.05$) by combing GO with KEGG analyses to build a ceRNA network.

## Construction of a lncRNA-related ceRNA network

According to the ceRNA hypothesis, lncRNA can regulate mRNA expression by sequestering and binding miRNAs like miRNA sponges. We constructed the ceRNA network through three steps: (1) we identified intersecting lncRNAs, miRNAs, and mRNAs that were differentially expressed in four stages (including up- and down-regulation); (2) we predicted lncRNA-miRNA interactions by miRanda (http://www.microrna.org/microrna/home.do) and used the Targetscan (http://www.targetscan.org) and miRanda databases to find target genes; and (3) we integrated aberrantly expressed data from the TCGA and the predicted miRNA information. Through these steps, we constructed a ceRNA network with intersecting lncRNAs and mRNAs that were negatively regulated by miRNAs. Additionally, a figure for the ceRNA network was depicted using Cytoscape v3.0.

## PPI network analysis

To clarify the potential relationships between the aberrantly expressed genes and RCC, we performed a protein-protein interaction (PPI) network analysis with the online software String (score >0.4).

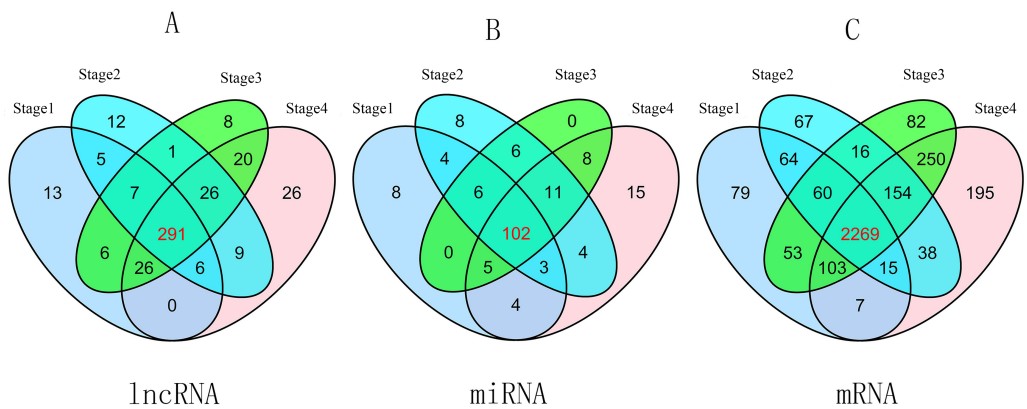

**Figure 2** **Venn diagram of aberrant expression profiles of lncRNAs (A), miRNAs (B) and mRNAs (C) between stage I RCC group (tumor tissues according to AJCC)/control group (non-tumor tissues), stage II group/control group, stage III group/control group, and stage IV group/control group.** A total of 291 intersecting lncRNAs, 102 intersecting miRNAs and 2,269 mRNAs were identified.

## Statistical analysis

Differentially expressed lncRNAs, miRNAs, and mRNAs were analyzed by BRB array tool software (https://brb.nci.nih.gov/BRB-ArrayTools/download.html). To identify the relationships between lncRNAs/miRNAs/mRNAs in the ceRNA network and to determine overall survival, we conducted univariate Cox proportional hazards regression and performed Kaplan–Meier survival analyses and log-rank tests to identify the relationships between RNA expression levels and survival time ($P < 0.05$ was considered statistically significant).

## RESULTS

### Cancer-specific lncRNAs in RCC

To compare stage I RCC tumor tissues and adjacent non-tumor RCC tissues from the TCGA database, we found 354 differentially expressed lncRNAs ($P \leq 0.01$, FDR $\leq 0.01$, FC $\geq 2$). Similarly, we identified 357 differentially expressed lncRNAs between stage II RCC tumor tissues and adjacent non-tumor RCC tissues. Three hundred eighty-five differentially expressed lncRNAs were identified between stage III RCC tumor tissues and non-tumor RCC tissues, and 404 differentially expressed lncRNAs were identified between stage IV RCC tumor tissues and non-tumor RCC tissues. Then, we found 291 intersecting lncRNAs from the above lncRNAs (92 up-regulated; 199 down-regulated) (Fig. 2A, Table S1). A heat map of these 291 aberrant lncRNAs was established using R software (Fig. S1), which revealed different expression patterns of lncRNAs in RCC.

### GO, KEGG pathway analyses and GSEA

A total of 2,650 mRNAs ($P \leq 0.01$, FDR $\leq 0.01$, FC $\geq 3$) were differentially expressed between stage I RCC tumors and non-tumor RCC tissues from the TCGA database, 2,683 mRNAs were differentially expressed between stage II RCC tumors and non-tumor RCC tissues, 2,987 mRNAs were differentially expressed between stage III RCC tumors and

non-tumor RCC tissues, and 3,031 mRNAs were differentially expressed between stage IV RCC tumors and non-tumor RCC tissues. Next, we identified 2,269 intersecting mRNAs (Fig. 2C) from the above mRNAs (1,132 up-regulated; 1,147 down-regulated).

We further explored the biological functions of these up- and down-regulated protein-coding RNAs by GO and KEGG pathway enrichment analyses. Using a GO database for comprehensive analysis, we found that the top three processes of the up-regulated mRNAs involved immune responses, signal transduction, and inflammatory responses, while the top three processes of the down-regulated mRNAs were transmembrane transport, small molecule metabolic processes, and excretion (Fig. 3A). The significant pathways of the up-regulated mRNAs included cytokine-cytokine receptor interactions, Staphylococcus aureus infection, and cell adhesion molecules, which were determined by KEGG pathway enrichment analysis (Fig. 3B). The pathways correlated with the down-regulated mRNAs included metabolic pathways, neuroactive ligand–receptor interactions, protein digestion, and absorption (Fig. 3B). In addition, GSEA of up and down-regulated mRNAs was implemented using the gene sets of GO and KEGG pathway database (Tables S2–S5). Finally, we found 738 significant intersecting mRNAs by GO and KEGG analyses ($P < 0.05$, FDR $< 0.05$).

## Construction of the ceRNA network

We found 132 RCC-related miRNAs that were differentially expressed between stage I tumors and non-tumor tissues ($P < 0.05$, FDR $< 0.05$, FC $\geq 2$). Additionally, 144 miRNAs were differentially expressed between stage II tumors and non-tumor tissues, 138 miRNAs were differentially expressed between stage III tumors and non-tumor tissues, and 152 miRNAs were differentially expressed between stage IV tumor and non-tumor tissues. Subsequently, we identified 102 differentially expressed miRNAs from the above four groups (41 up-regulated; 61 down-regulated) (Fig. 2B).

Integrating 291 intersecting lncRNAs and 102 intersecting miRNAs with those that we identified from the prolife TCGA data, we finally chose 106 lncRNAs that were negatively related to 26 miRNAs that were predicted by the miRanda database. These 26 miRNAs were negatively related to 69 mRNAs following a comparison of data predicted by Targetscan and miRanda with the data from GO and KEGG analyses. Based on these data, we established a miRNA-lncRNA-mRNA ceRNA network (Fig. 4).

## Construction of the PPI

To better understandthe critical genes predicted in RCC, we established a PPI network that included 44 genes with scores of >0.4 in String (Fig. 5). From the figure, we determined that the main hub nodes were VEGFA, NTRK2, DLG2, E2F2, MYB and RUNX1. Hence, the key genes associated with RCC can be predicted by our network.

## RNAs in the ceRNA network are related to survival

To identify the miRNAs associated with prognosis, all miRNAs in the ceRNA network were analyzed via univariate Cox proportional hazards regression, and four miRNAs (miR-9-5p, miR-21-5p, miR-155-5p, and miR-244-5p) were observed to be significantly changed (log-rank $P < 0.05$). Additionally, we found 63 lncRNAs and 31 mRNAs that were

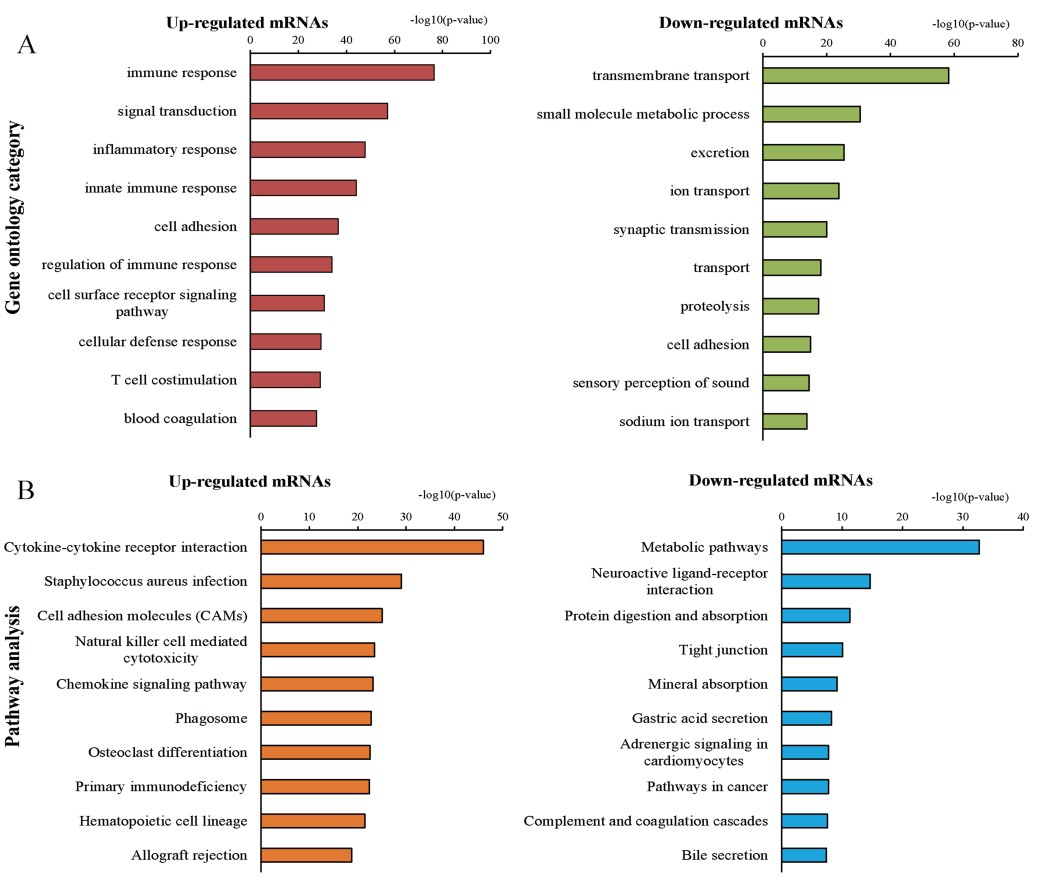

**Figure 3** **Gene ontology (GO) and Kyoto Encyclopedia of Genes and Genomes (KEGG) pathway analyses.** (A) Top 10 processes of up- and down-regulated genes in the GO analysis. (B) Top 10 pathways of up- and down-regulated genes in the pathway analysis.

significantly related to overall survival (Table S6). Here, we show figures for RNAs (Fig. 6), as the coexpression profiles of RNAs in the lncRNA-miRNA-mRNA correlations were all significantly associated with overall survival. These relationships are shown in Table 1.

# DISCUSSION

LncRNAs are emerging as having crucial roles in various cell biological processes and diverse malignant tumors (*Martens-Uzunova et al., 2014*; *Schmitt & Chang, 2016*). Although their exact mechanisms are unclear, a variety of theories and hypotheses have been described regarding how lncRNAs participate in regulating malignant biological behavior. Some studies have documented that lncRNAs can affect cancer-related mRNAs by interacting with miRNAs, yielding new perspectives regarding lncRNA functions (*Lv et al., 2016*; *Yue et al., 2016*). This hypothesis has been supported *in vivo* and *in vitro*; moreover, ceRNA coexpression networks have been established for several human tumors (*Li et al., 2016a*; *Li et al., 2016b*; *Liu et al., 2016a*; *Liu et al., 2016b*; *Wang et al., 2016a*; *Wang et al., 2016b*; *Yang et al., 2016*). A few studies have reported interactions between lncRNAs and miRNAs in

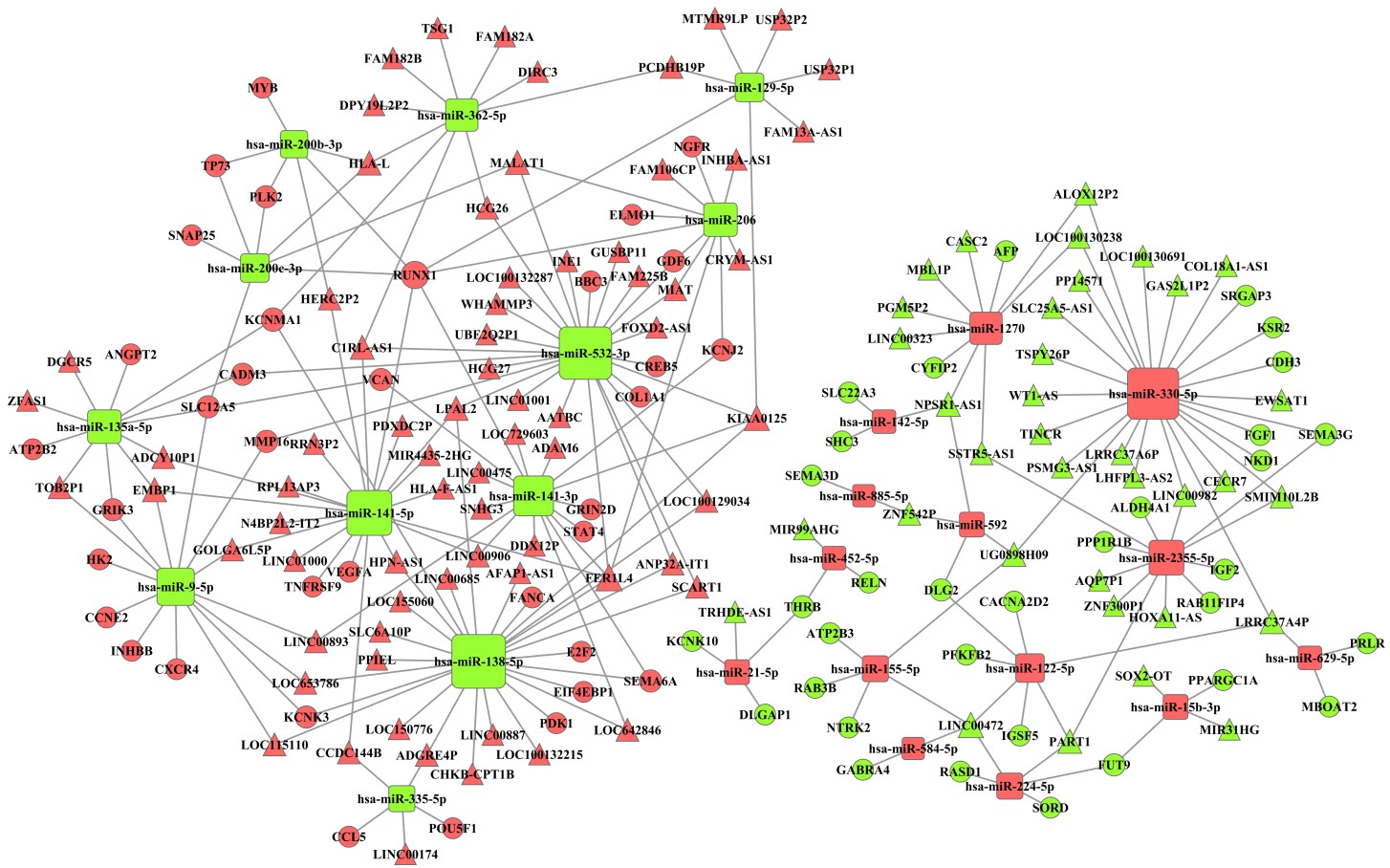

**Figure 4** **The miRNA-lncRNA-mRNA ceRNA network is composed of 106 lncRNAs, 26 miRNAs and 69 mRNAs.** Red diamonds represent up-regulated miRNAs, while green diamonds represent down-regulated miRNAs. Triangles and circles represent lncRNAs and mRNAs, respectively, while red indicates up-regulation, and green indicates down-regulation.

RCC. Furthermore, one previous study suggested that some lncRNAs are associated with chromophobe RCC and that they may function as ceRNAs in complex ceRNA networks (*He et al., 2016*). Nevertheless, coexpression networks based on large-scale bioinformatics data and the potential roles of lncRNAs in ceRNA networks are still poorly understood in RCC. In the present study, we confirmed aberrantly expressed lncRNAs, miRNAs and mRNAs based on the RNA expression profiles of 457 RCC patient tissue samples from the TCGA. Additionally, we constructed a ceRNA network with cancer-specific lncRNAs, miRNAs and mRNAs. To understand the systematic biological roles of the abnormally expressed RNAs, we further investigated the association between these RNAs and overall survival, and we also constructed a PPI network to find key mRNAs.

After analyzing the RNA sequencing data from the TCGA, we finally identified 291 cancer-specific lncRNAs that were abnormally expressed in different stages of RCC. We then constructed a cluster analysis map to show the differential distributions of the 291 lncRNAs in RCC tumor tissues and adjacent non-tumor tissues. Among these lncRNAs, several known lncRNAs, such as GAS5, CASC2, TCL6, MALAT-1, UCA1 and HOTAIR,

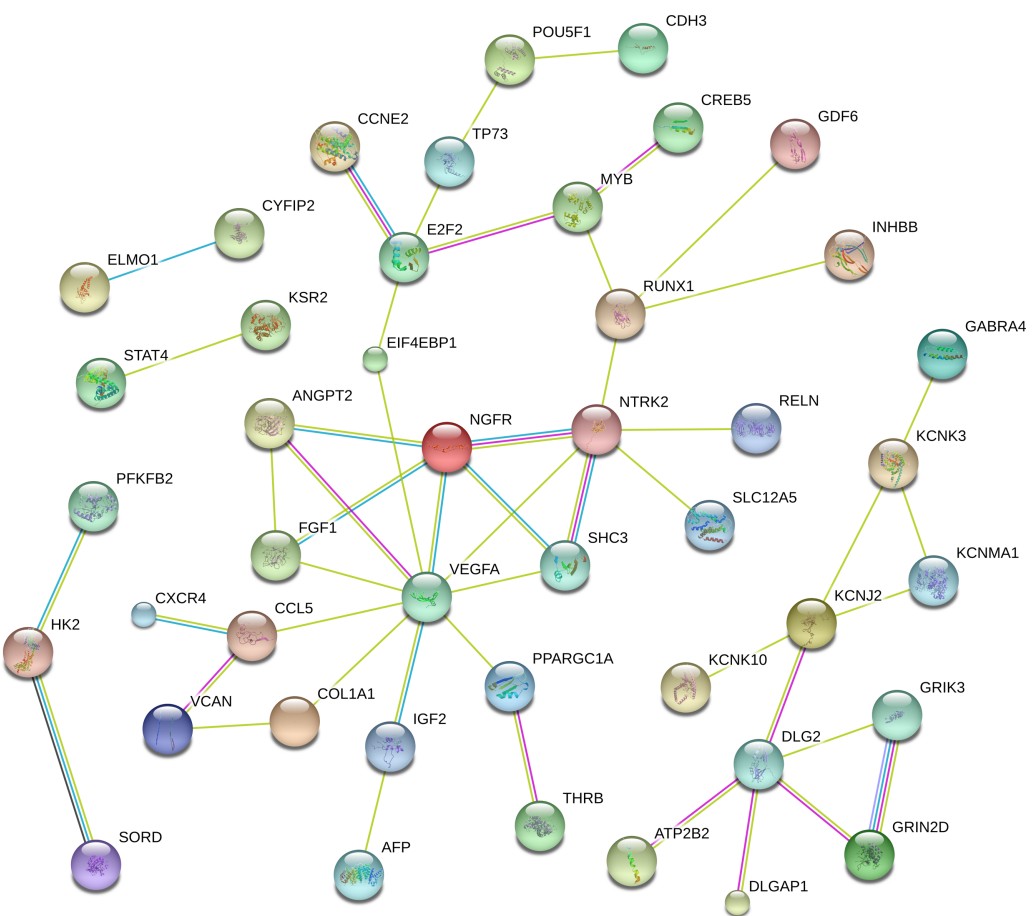

**Figure 5** **The protein-protein interaction network included 44 genes with scores of > 0.4 in String.**

have been found to be related to RCC cell proliferation, invasion and apoptosis, and they are important regulators in RCC occurrence and progression (*Cao et al., 2016*; *Chen et al., 2017*; *Hong et al., 2017*; *Li et al., 2016a*; *Li et al., 2016b*; *Qiao et al., 2013*; *Su et al., 2017*). Furthermore, the massive RNA sequencing data identified in the present study revealed hundreds of differentially expressed lncRNAs that have not yet been reported and may function as novel oncogenes and theranostic markers in RCC.

The GO and KEGG pathway analyses of the 2,269 the intersecting mRNAs revealed the main biological processes and pathways in RCC. Many of them were classical pathways comprising important areas of RCC research, such as immune responses, signal transduction, and metabolism (*Labrousse-Arias et al., 2017*; *Sakai, Miyake & Fujisawa, 2013*; *Wettersten et al., 2017*). However, some of biological processes and pathways, such as protein digestion and absorption, were reported in other malignancies (*Dong et al., 2017*; *Li et al., 2017*). From our enrichment analysis of large-scale samples focusing on RCC, we discovered multiple processes involved in neoplasm formation, which was supplementary to the results of previous studies. To enhance the validity and reliability of the ceRNA

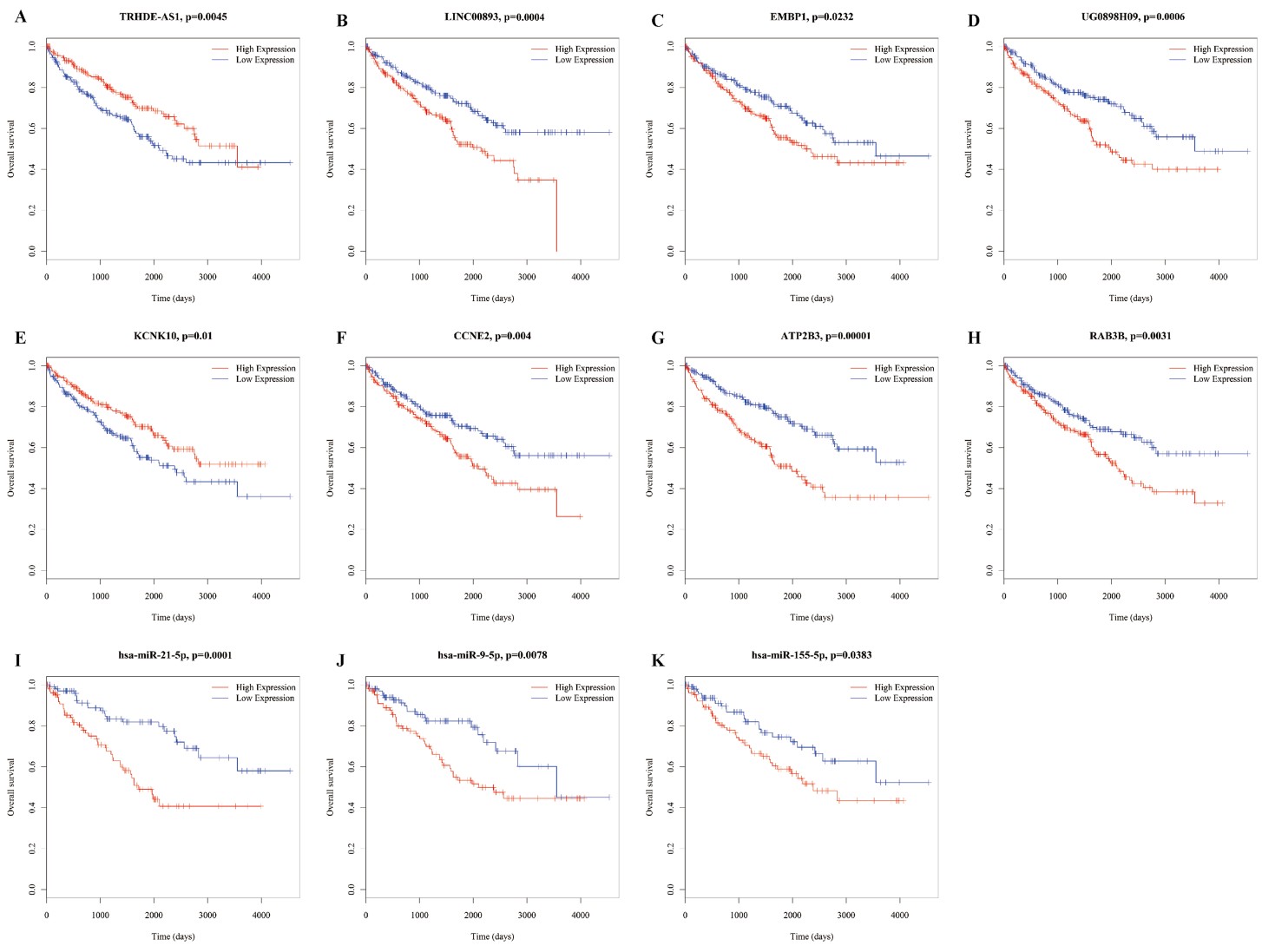

**Figure 6** **Kaplan–Meier survival curves for 11 related RNAs (4 lncRNAs, 3 miRNAs and 4 mRNAs) in the ceRNA network (horizontal axis: overall survival times: days, vertical axis: survival function).** High expression of LINC00893 (B), EMBP1 (C), UG0898H09 (D), CCNE2 (F), ATP2B3 (G), RAB3B (H), has-miR-21-5p (I), has-miR-5p (J) and has-miR-155-5p (K) is associated with low proportion of overall survival ($P = 0.0004$, 0.0232, 0.0006, 0.004, 00001, 0.0031,0.0001, 0.0078, 0.0383, respectively); high expression of TRHDE-AS1 (A) and KCNK10 (E) is associated with high proportion of overall survival ($P = 0.0045$ and 0.001, respectively).

network, we selected the mRNAs that significantly participatedin both the GO and KEGG pathways for further analyses.

To provide a comprehensive view on how these lncRNAs participate in multiple biological processes, we constructed a ceRNA network combining the intersecting miRNAs and mRNAs, as many studies have elucidated the possibility that lncRNAs can be ceRNA regulators andcan communicate with other RNA transcripts (*Lv et al., 2016*; *Qu et al., 2016*; *Salmena et al., 2011*; *Xiao et al., 2015*; *Yue et al., 2016*). To increase the accuracy of the ceRNA network, we integrated the intersecting RNAs in different stages of RCC

**Table 1** RNAs significantly associated with overall survival in lncRNA-miRNA-mRNA pathway.

| lncRNA | miRNA | mRNA |
|--------|-------|------|
| TRHDE-AS | miR-21-5p | KCNK10 |
| Lin00893 | miR-9-5p | CCNE2 |
| EMBP1 | miR-9-5p | CCNE2 |
| UG0898H09 | miR-155-5p | ATP2B3 |
| UG0898H09 | miR-155-5p | PAB3B |

and evaluated the relationships between lncRNAs and miRNAs and between miRNAs and mRNAsthat were predicted by the Targetscan and miRanda databases. Ultimately, 106 lncRNAs, 26 miRNAs and 69 mRNAs were involved in the ceRNA network. Recent studies have demonstrated that down-regulation of the lncRNA CASC2 by miRNA-21 increases RCC cell proliferation and migration (*Cao et al., 2016*) and that MALT1 promotes aggressive RCC through Ezh2 and interacts with miR-205 (*Hirata et al., 2015*). The key lncRNAs identified above were also found in our ceRNA network. Additionally, our study integratively predicted the potential relationships between novel lncRNAs, miRNAs and mRNAs in RCC. Construction of a PPI network increases the reliability of our ceRNA network in the following ways: First, some genes described as main hub nodes, such as VEGFA, NTRK2, DLG2, E2F2, MYB and RUNX1, are themselves vital genes in human tumors, and some of them are directly associated with RCC (*Chimge et al., 2016*; *Gao et al., 2016*; *Gonda & Ramsay, 2016*; *Jones et al., 2013*; *Ma et al., 2015*; *Zubakov, Stupar & Kovacs, 2006*). Second, some genes in the ceRNA network that have not been reported to be correlated with RCC may also be oncogenes, as they can interact with RCC-related genes. The reasons above validate the hypotheses that the genes in the ceRNA network caninfluence RCC and that lncRNAs can be regulators that modulate oncogene expression.

To further evaluate the roles of RNAs in RCC diagnosis and prognosis, we analyzed the associations between RNAs in the ceRNA network and patient survival. We found four miRNAs, 63 lncRNAs and 31 mRNAs that were significantly correlated with overall RCC survival. Up-regulation of the lncRNA MALAT1 has been reported to correlate with tumor progression and poor prognosis in RCC (*Zhang et al., 2015*), while some lncRNAs have been reported in other tumors, such as HOXA11-AS, which was regarded as a biomarker of poor progression in glioma (*Wang et al., 2016a*; *Wang et al., 2016b*). However, most of the 19 lncRNAs have not yet been reported. Subsequently, we found some RNAs in the lncRNA-miRNA-mRNA correlations that were uniformly related to overall survival, such as miR-21-5p-regulated mRNA KCNK10 and lncRNA TRHDE-AS1, which confirmed the internal relationships constructed in the ceRNA network in the present study.

Our work identified many valuable RNAs that are differentially expressed in RCC tissues via bioinformatics analysis. Some of these RNAs have been verified *in vivo* and *in vitro* experiments. However, most of the aberrant RNAs still need to be validated, and our ceRNA network, which was constructed *in silico*, needs to be validated with additional biological experiments.

## CONCLUSION

The present study successfully identified hundreds of differentially expressed lncRNAs, miRNAs and mRNAs in RCC by bioinformatics analysis from candidate data from the TCGA. Moreover, we determined the biological processes and pathways via GO and KEGG pathway analyses with cancer-specific mRNAs in RCC. Importantly, we constructed a ceRNA network to explore the potential roles of lncRNAs in RCC, which can serve as a reference for further research. We also investigated the associations between RNAs and overall survival and found that some of the RNAs could be used as biomarkers for RCC diagnosis and prognosis.

## ACKNOWLEDGEMENTS

We thank The Cancer Genome Atlas (TCGA) project and its contributors for this valuable public data set.

### Funding
The authors received no funding for this work.

### Competing Interests
The authors declare there are no competing interests.

### Author Contributions

- Qianwei Xing conceived and designed the experiments, performed the experiments, analyzed the data, contributed reagents/materials/analysis tools, prepared figures and/or tables.
- Yeqing Huang, You Wu and Limin Ma performed the experiments, contributed reagents/materials/analysis tools, prepared figures and/or tables.
- Bo Cai conceived and designed the experiments, contributed reagents/materials/analysis tools, authored or reviewed drafts of the paper, approved the final draft.

### Data Availability
BO CAI. (2018). TCGA-RCC [Data set]. Zenodo. DOI 10.5281/zenodo.1293051.

### Supplemental Information
Supplemental information for this article can be found online at http://dx.doi.org/10.7717/peerj.5124#supplemental-information.

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
