# Peer review of "Integrated analysis of differentially expressed profiles and construction of a competing endogenous long non-coding RNA network in renal cell carcinoma"

_PeerJ, doi:10.7717/peerj.5124_

## Round 0.1 · original submission · Major Revisions

As raised by two reviewers, the author need to explain the statistical methods in more detail. Because this is the foundation for all the follow-up analysis, we suggest to re-do those student t-test for identifying differentially expressed lncRNAs, miRNAs, and mRNAs if necessary.

Reviewer 1 ·

Basic reporting

no comment

Experimental design

no comment

Validity of the findings

This paper may not be considered as statistically sound. Student t-test may not be a suitable method for identifying differentially expressed lncRNAs, miRNAs, and mRNAs.

Additional comments

The authors have identified cancer- specific lncRNAs and constructed a ceRNA network for RCC. A survival analysis related to the RNAs revealed candidate biomarkers for further study in RCC. However, the student t-test used here may be problematic in identifying differentially expressed lncRNAs, miRNAs, and mRNAs. This major issue must be addressed and solved before acceptance.

Major revisions:

1). In the “Statistical analysis” section, the authors used Student’s t test to find differentially expressed lncRNAs, miRNAs, and mRNAs, this approach may be problematic. As we know student t-test only can be apply to those data which follows normal distribution, it cannot be applied to those expression counts since those counts are most likely follow negative-binomial distribution. That’s why some DEG tools, e.g. DESeq and edgeR are more accurate in identifying DEGs in RNA-seq data. In my opinion, even we do not really sure about the distribution of these data, then a non-parametric method, e.g. Wilcoxon Rank-Sum Test will be more suitable and safer to use in this case. The authors need to give solid explanations that why they can use student t-test here, otherwise the differentially expressed lncRNAs, miRNAs, and mRNAs which obtained in this step will be very questionable for the downstream analysis.

2). In lines 120 and 121, could authors explain what’s the reason for choosing these different cutoffs of p-value, FDR, and FC for lncRNAs, miRNAs, and mRNA? I suggest to give people more details in the method section.


Minor revisions:

line 87, lincRNA -> lncRNA

For the titles of subsections of “Materials and methods”, e.g. “Patients and samples”, the space in the front may need to be removed

Line 99, remove “inclusion”

Reviewer 2 ·

Basic reporting

no comment

Experimental design

no comment

Validity of the findings

no comment

Additional comments

The manuscript from Qianwei Xing, and colleagues used existing sequencing data from TCGA to investigate the differential expression profiles of a cancer-specific lncRNA and ceRNA coexpression network in RCC, and their work would be very useful for elucidate the functions of lncRNAs in RCC. Overall, I think the authors supplied useful approach and theoretical basis of mechanisms for RCC research. Besides, I think it is a good idea to investigate data on public database and make full use of the existing data.

I have several minor concerns about this paper.
1. Are the p values in line 118 are adjust p valuse? Is yes, please mark them clearly; if not, please clarify what kind of p value they are.
2. I think it would be better to add GSEA analysis to this study.
3. The resolution of figures need to be improved, especial fig2 and fig6.
4. Paragraph transitions need to improve in the introduction part
5. I suggest to add reads or counts data used in this paper as a supplemental table.

---

## Round 0.2 · Minor Revisions

Please add quality control information as suggested by reviewers, which is important for your data quality.

Reviewer 1 ·

Basic reporting

no comment

Experimental design

no comment

Validity of the findings

no comment

Additional comments

I am satisfied with the revision. I agree with reviewer 2 about adding GSEA results to this paper to make it more scientific solid. GSEA is a software of gene set enrichment analysis, not a database similar to TCGA. It should be relatively easy to add it to this paper.

Reviewer 2 ·

Basic reporting

All the text in figures should be the same font, please check carefully and modify.
The data to show the quality of the libraries used in this paper should be added.

Experimental design

no comment

Validity of the findings

no comment

Additional comments

The revised manuscript is much better than previously one. The introduction and methods were improved and more detail were added. I am satisfied with the replies to all my concerns except one. The reason why I suggested to add raw counts data was to show the quality of the libraries. A very simple table with total reads, mapping rate should be totally fine. Good libraries are very import base for accurate analysis. So I insist to add QC of libraries used in this paper.

After go through carefully of the revised manuscript, I still find some minor problems, such as line 173 "of2,650", a space should be added between "of" and "2,650". One more thing, the text in figures should be the same font. I suggest author check the manuscript carefully and try your best to reach all the publication standards of Peer J.

---

## Round 0.3 · accepted · Accept

Thank you for your choosing PeerJ, I am satisfied for your revision.

#